# PA28γ promotes the malignant progression of tumor by elevating mitochondrial function via C1QBP

**Jiongke Wang**[1†], **Yujie Shi**[1,2†], **Ying Wang**[1], **Yingqiang Shen**[1], **Huan Liu**[3], **Silu Sun**[1], **Yimei Wang**[1], **Xikun Zhou**[3], **Yu Zhou**[1], **Xin Zeng**[1*], **Jing Li**[1*], **Qianming Chen**[1]

[1]State Key Laboratory of Oral Diseases & National Center for Stomatology & National Clinical Research Center for Oral Diseases & Research Unit of Oral Carcinogenesis and Management & Chinese Academy of Medical Sciences, West China Hospital of Stomatology, Sichuan University, Chengdu, China; [2]Department of Stomatology, The First Affiliated Hospital with Nanjing Medical University, Nanjing, China; [3]State Key Laboratory of Biotherapy and Cancer Center, West China Hospital, Sichuan University and Collaborative Innovation Center for Biotherapy, Chengdu, China

**\*For correspondence:**
zengxin22@163.com (XZ);
lijing1984@scu.edu.cn (JL)

†These authors contributed equally to this work

## eLife Assessment

This manuscript determines how PA28g, a proteasome regulator that is overexpressed in tumors, and C1QBP, a mitochondrial protein for maintaining oxidative phosphorylation that plays a role in tumor progression, interact in tumor cells to promote their growth, migration and invasion. Additional experiments and analyses that supported the theoretical models for the interaction have been performed in response to the reviews. The overall findings and conceptual framework are **important** and the evidence is **solid**. A logical extrapolation of this work is to test the C1QBP mutants using functional assays to determine whether the mutations can decrease the protein stability mediated by the interaction with PA28g.

**Abstract** Proteasome activator 28γ (PA28γ) plays a critical role in malignant progression of various tumors, however, its role and regulation are not well understood. Here, using oral squamous cell carcinoma (OSCC) as the main research model, and combining co-immunoprecipitation (Co-IP), proximity ligation assays (PLA), AlphaFold 3-based molecular docking, and truncation constructs, we discovered that PA28γ interacted with complement 1q binding protein (C1QBP). This interaction is dependent on the C1QBP N-terminus (aa 1–167) rather than the known functional domain. Point mutation in C1QBP (T76A/G78N) disrupting predicted hydrogen bonding with PA28γ-D177 significantly reduced their binding. Notably, we found that PA28γ enhances C1QBP protein stability in OSCC. Functionally, PA28γ and C1QBP co-localized in mitochondria, promoting fusion (via upregulation of OPA1, MFN1/2), respiratory complex expression, oxidative phosphorylation (OXPHOS), ATP production, and ROS generation. Crucially, PA28γ-enhanced OSCC cell migration, invasion, and proliferation in vitro were dependent on C1QBP. In vivo, orthotopic OSCC models showed Pa28γ overexpression increased tumor growth and elevated C1qbp levels, correlating with elevated ATP and ROS. Using transgenic *Psme3*[-/-] mice and subcutaneous tumor grafts, we confirmed that silencing of Pa28γ suppresses tumor growth, reduces C1qbp levels, and dampens mitochondrial metabolism—specifically in knockout hosts. Clinically, PA28γ and C1QBP expression were positively correlated during oral carcinogenesis and in metastatic OSCC tissues across cohorts. High co-expression predicted poor prognosis in OSCC patients. Thus, PA28γ stabilizes C1QBP via N-terminal interaction to drive mitochondrial OXPHOS and tumor progression, highlighting its potential as a therapeutic target.

## Introduction

PA28γ, also known as REGγ or *PSME3*, is a well-known non-ATP-dependent proteasome regulator. PA28γ is overexpressed in several malignant tumors, including thyroid, colon, and breast cancer, and is significantly associated with poor prognosis (*Mao et al., 2008*; *Stadtmueller and Hill, 2011*). By regulating the stability of key components in various complex signaling pathways, PA28γ influences the biological behavior of tumor cells. For instance, PA28γ interacts with both MDM2 and p53 proteins and facilitates their physical interaction, which promotes ubiquitination and MDM2-dependent proteasomal degradation of p53 and inhibits apoptosis after DNA damage (*Zhang and Zhang, 2008*). Our previous studies revealed that the expression level of PA28γ in OSCC cancer nest tissues is positively correlated with patient prognosis (*Li et al., 2015*; *Liu et al., 2018*). However, its role and regulation are not well understood.

OSCC is the most common oral and maxillofacial malignant tumor (*Siegel et al., 2023*; *Sung et al., 2021*). The pathogenic factors of OSCC include smoking, alcohol consumption, and viral infections, such as human papillomavirus (HPV). Therapies for OSCC include surgery, radiotherapy, chemotherapy, and immunotherapy. Despite the rapid development of imaging, surgery, radiotherapy, and immunotherapy in recent years, along with the emphasis on personalized treatment and multidisciplinary cooperation in the treatment of OSCC, the overall survival rate of OSCC patients has not significantly improved (*Peres et al., 2019*). The main reasons for unsatisfactory treatment efficacy include invasion of surrounding tissue, early lymph node metastasis, and a tendency toward recurrence (*Lindemann et al., 2018*). Metabolic reprogramming is a crucial characteristic of tumors, closely associated with malignant behaviors such as tumor growth, invasion, metastasis, and immune escape (*Ohshima and Morii, 2021*). Current research on metabolic reprogramming in OSCC primarily focused on the mechanism of glycolytic metabolism and metabolic shift from glycolysis to oxidative phosphorylation (OXPHOS) of oral squamous cell carcinoma, which lays the groundwork for novel therapeutic interventions to counteract OSCC (*Chen et al., 2024*; *Zhang et al., 2020*). Understanding the molecular events that lead to the occurrence and development of OSCC can aid in developing new tumor-targeted therapies, which hold significant clinical application potential.

The complement 1q binding protein (C1QBP) is a crucial protein for maintaining mitochondrial function, particularly in mitochondrial OXPHOS (*Ghebrehiwet et al., 2019*; *Wang et al., 2022*). OXPHOS is essential for the production of adenosine triphosphate (ATP), which is necessary for tumor development. C1QBP has been shown to play a significant role in cancer progression, as influencing tumor growth, invasion, and metastasis (*Bai et al., 2019*; *Hou et al., 2022*; *Vendramin et al., 2018*). Due to these roles, C1QBP presents a promising therapeutic target for various tumors, including melanoma, breast cancer, and colorectal cancer (*Matsumoto and Bay, 2021*). In OSCC, enhanced mitochondrial OXPHOS function might also be linked to malignant tumor progression (*Vyas et al., 2021*; *Xiao et al., 2021*; *Zhu et al., 2021*), although the underlying mechanism remains unclear.

In this study, we discovered that PA28γ interacts with C1QBP and that PA28γ can stabilize C1QBP. This interaction enhances OXPHOS and promotes the growth, migration, and invasion of OSCC cells. Additionally, the expression of PA28γ and C1QBP is increased and positively correlated in OSCC. Therefore, PA28γ and C1QBP are potential targets for the treatment and prognosis of cancer. In our knowledge, it is the first study to describe the undiscovered role of PA28γ in promoting the malignant progression of OSCC by elevating mitochondrial function, providing new clinical insights for the treatment of OSCC.

## Results

### PA28γ interacts with and stabilizes C1QBP

To explore how PA28γ promotes OSCC progression, we conducted a gene–gene interaction analysis using the GeneMANIA database. This analysis identified *C1QBP* as a significant gene within the interaction network, with the protein-coding gene of *PSME3* also playing a key role in the *C1QBP* interaction network (*Figure 1—figure supplement 1A, B*). Subsequently, pull-down and coimmunoprecipitation analysis demonstrated that both endogenous and exogenous PA28γ and C1QBP could bind to each other (*Figure 1A, B*; *Figure 1—figure supplement 1C, D*). Moreover, proximity ligation assay (PLA) revealed a close proximity between PA28γ and C1QBP in cells, thereby suggesting a potential interaction between the two proteins (*Figure 1C*). Then, we developed four OSCC cell lines

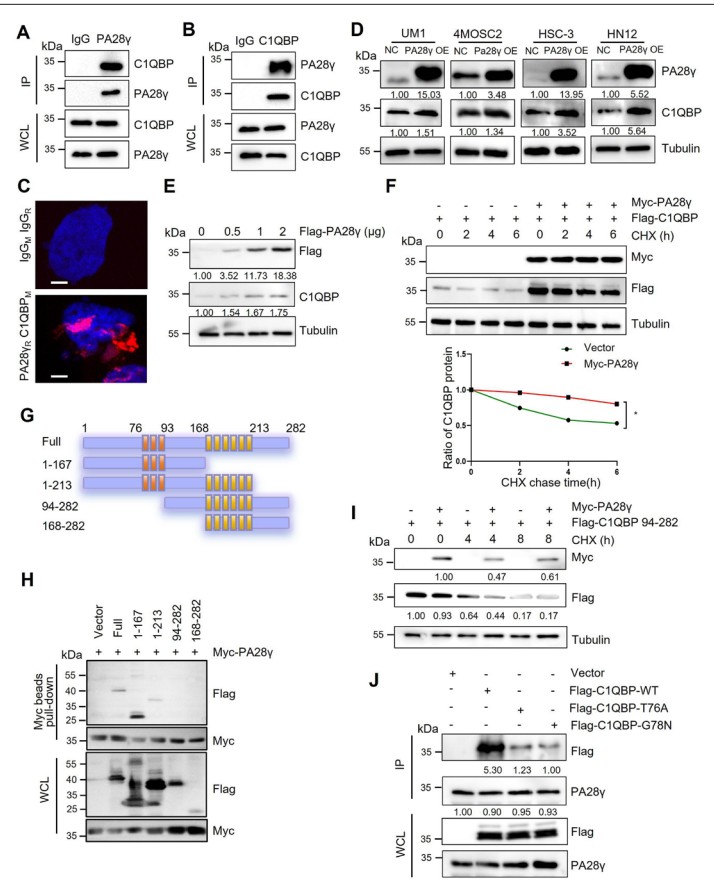

**Figure 1.** The interaction between proteasome activator 28γ (PA28γ) and C1QBP. (**A**, **B**) The interaction between endogenous PA28γ and C1QBP in HSC-3 cells was verified via immunoprecipitation (IP). (**C**) Proximity ligation assay (PLA) image of UM1 cells shows the interaction between C1QBP and PA28γ in both cytoplasm and nucleus (red fluorescence, scale bar=5μm). (**D**) Western blot analysis of C1QBP in four oral squamous cell carcinoma (OSCC) cell lines with PA28γ overexpression. (**E**) Western blot analysis of C1QBP in 293T cells transfected with increasing doses of Flag-PA28γ. (**F**) 293T cells transfected with Flag-C1QBP with or without Myc-PA28γ were treated with cycloheximide (CHX) (100 μg/ml) for the indicated periods of time. Quantification of Flag-C1QBP levels relative to tubulin levels is shown (the data are representative of one experiment with three independent biological replicates, *p<0.05). (**G**) Full-length C1QBP and truncation with deletion of functional domains. (**H**) Pull-down of 293T cells transfected with Myc-PA28γ and full-length Flag-C1QBP or truncation mutants of functional domains for 36 hr. (**I**) 293T cells transfected with Flag-C1QBP 94-282 with or without Myc-PA28γ were treated with CHX (100 μg/ml) for the indicated periods of time. (**J**) IP of 293T cells transfected with Flag-C1QBP wild-type or mutations (T76A and G78N) for 48 hr.

The online version of this article includes the following figure supplement(s) for figure 1:

**Figure supplement 1.** The interaction between proteasome activator 28γ (PA28γ) and C1QBP.

with stable overexpression of PA28γ using a Flag-PA28γ-expressing lentivirus and observed that the protein level of C1QBP was higher in these cell lines compared to control cells (*Figure 1D*). Notably, PA28γ upregulated C1QBP protein levels in a dose-dependent manner (*Figure 1E*). To determine whether the upregulated C1QBP was due to increased stability, cells were pretreated with cyclo-heximide (CHX) to inhibit protein synthesis. As a result, in the presence of PA28γ, C1QBP exhibited a significantly lower turnover rate compared to the control group (*Figure 1F*). In addition, PA28γ stabilized C1QBP in the absence and presence of a proteasome inhibitor MG-132 (*Figure 1—figure supplement 1E*), whereas the inhibitor alone did not stabilize C1QBP. Furthermore, AlphaFold 3-based structural modeling predicted that the PA28γ-C1QBP interaction could be mediated by the N-terminal region of C1QBP (amino acid residues 1–167), rather than the known functional domain (amino acids 168–213) that bind to mitochondrial antiviral proteins (*Peerschke et al., 2020*; *Xu et al.,*

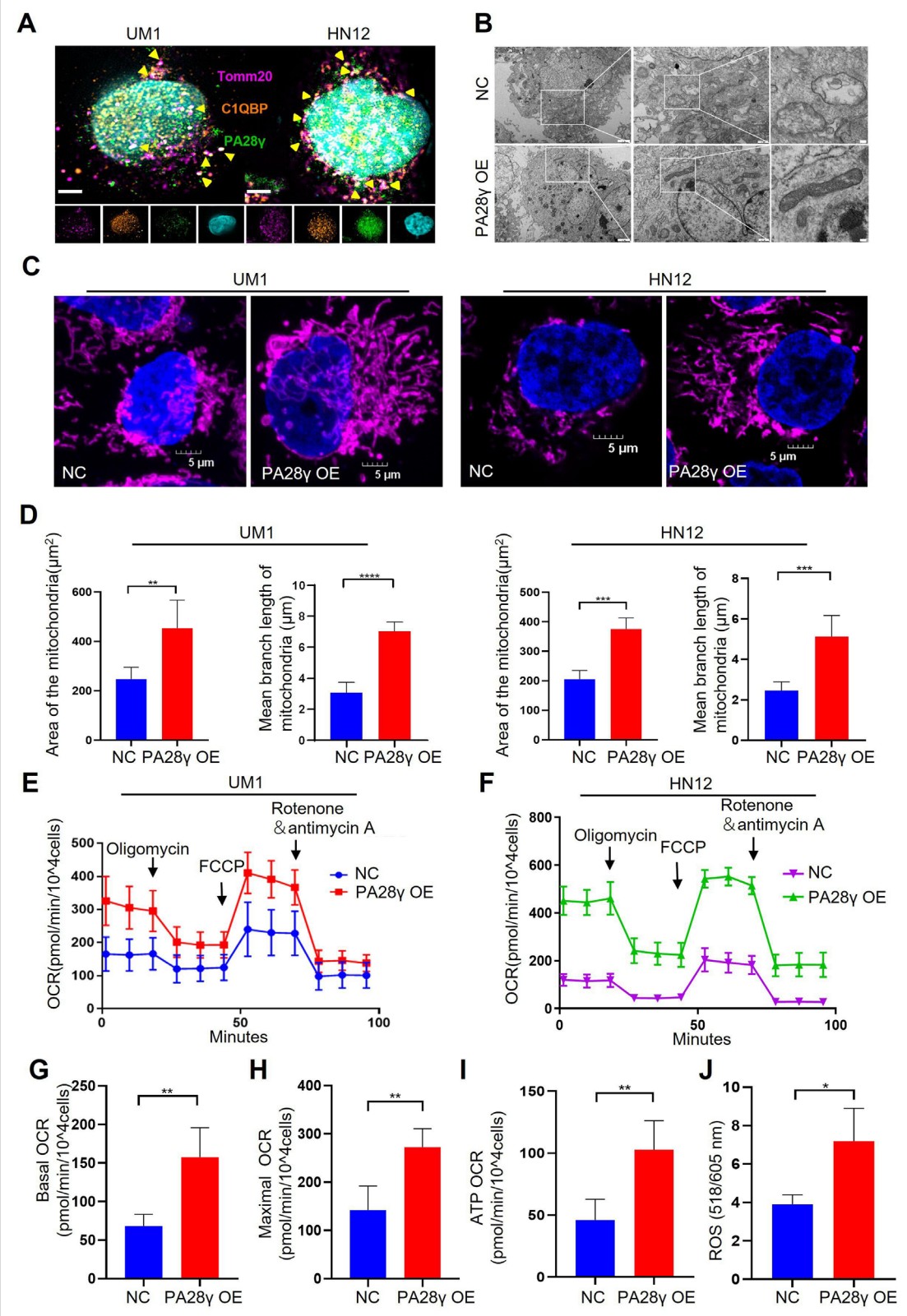

**Figure 2.** Proteasome activator 28γ (PA28γ) and C1QBP colocalize in mitochondria and influence mitochondrial functions in vitro. (**A**) Confocal image of immunofluorescence (IF) in two oral squamous cell carcinoma (OSCC) cell lines (scale bar=5μm). (**B**) Transmission electron microscopy (TEM) images of PA28γ-overexpressing and control UM1 cells (scale bar=2μm). (**C**) Representative confocal images of mitochondria in two OSCC cell lines (scale bar=5μm). (**D**) The area and mean branch length of mitochondria in two OSCC cell lines were measured by ImageJ (the data are presented as the mean

*Figure 2 continued on next page*

*Figure 2 continued*

± SD of three independent experiments; **p<0.01, ***p<0.001, and ****p<0.0001). (**E**, **F**) Oxygen consumption rates (OCRs) of PA28γ-overexpressing and control UM1 and HN12 cells were plotted using a Cell Mito Stress Test Kit (the data are presented as the means ± SDs of three independent experiments). (**G**-**I**) Basal OCRs, maximal OCRs, and ATP production of PA28γ-overexpressing and control UM1 cells measured by the Cell Mito Stress Test (the data are presented as the means ± SDs of three independent experiments, **p<0.01). (**J**) ROS generation in PA28γ-overexpressing and control UM1 cells (the data are presented as the means ± SDs of three independent experiments; *p<0.05).

The online version of this article includes the following figure supplement(s) for figure 2:

**Figure supplement 1.** Proteasome activator 28γ (PA28γ) and C1QBP colocalize in mitochondria and regulate OXPHOS in oral squamous cell carcinoma (OSCC) cells.

*2009*; *Figure 1—figure supplement 1F-I*). Results from a pull-down analysis of the PA28γ interaction with full-length and truncated C1QBP proteins are consistent with this prediction (*Figure 1G-I*). In addition, we performed energy minimization to refine the structural model for PA28γ-C1QBP complex, and used the refined model to reveal potential interactions between the two proteins. The results suggest that the T76 and G78 residues in C1QBP would form hydrogen bonds with the D177 residue in PA28γ (*Figure 1—figure supplement 1J*). Consistently, co-immunoprecipitation analysis demonstrated that mutations that would disrupt these hydrogen bonds (T76A and G78N) significantly reduced the binding of C1QBP to PA28γ (*Figure 1J*). Therefore, our results illustrated that PA28γ could interact with C1QBP, in a manner dependent on the N-terminus of C1QBP, and that PA28γ could stabilize C1QBP.

## PA28γ and C1QBP colocalize in mitochondria and affect mitochondrial functions

To further investigate the physical interaction between PA28γ and C1QBP, we conducted an immunofluorescence (IF) assay in two OSCC cell lines, as shown, these two proteins were colocalized in the mitochondria (*Figure 2A*). Remarkably, transmission electron microscopy (TEM) images revealed fewer mitochondrial vacuoles and higher mitochondrial ridge density in PA28γ-overexpressing cells compared to control cells (*Figure 2B*, *Figure 2—figure supplement 1A*). IF analysis also showed increased mitochondrial lengths and areas in the PA28γ-overexpressing cells (*Figure 2C, D*). Following that, we conducted a Cell Mito Stress Test to measure the oxygen consumption rate (OCR) (*Figure 2E, F*). The results indicated significantly higher basal respiration, maximal OCRs, and ATP production in PA28γ-overexpressing cells compared to control cells (*Figure 2G-I*; *Figure 2—figure supplement 1B-D*). Considering that mitochondria are the primary site for ROS generation and that ROS play a crucial role in carcinogenesis, we measured ROS levels in OSCC cells. Consistently, ROS levels were significantly higher in PA28γ-overexpressing cells (*Figure 2J*, *Figure 2—figure supplement 1E*). In addition, experiments conducted in PA28γ-sh OSCC cells yielded opposite results, further confirming these conclusions (*Figure 2—figure supplement 1F-K*).

To demonstrate the carcinogenic abilities of PA28γ and C1QBP, we injected 4MOSC2 cells, a mouse OSCC line, with or without Pa28γ overexpression under the surface of the tongue in C57BL/6 mice (*Figure 3—figure supplement 1A*). After 10 days of normal feeding, the tumors in the Pa28γ-overexpressing groups were significantly larger than those in the control group. Furthermore, via immunohistochemical (IHC) staining revealed that PA28γ overexpression upregulates C1QBP in vivo (*Figure 3A-C*). Consistent with the in vitro experiments, ATP production and ROS generation were significantly increased (*Figure 3D, E*). Additionally, analysis of OSCC xenograft tumor tissue in nude mice revealed that PA28γ's regulation of C1QBP in OSCC cells is independent of the immune system (*Figure 3—figure supplement 1B, C*). In addition, we established stable Pa28γ-silenced B16 cells, a mouse melanoma cell line, and found that the level of C1qbp protein was decreased in Pa28γ-silenced B16 cells (*Figure 3—figure supplement 1D*). Cells with or without Pa28γ silencing were grafted into the flanks of *Psme3*[-/-] and *Psme3*[+/+] C57BL/6 mice by subcutaneous injection (*Figure 3—figure supplement 1E*). Remarkably, the tumors derived from the silenced Pa28γ cells in *Psme3*[-/-] mice were smaller than those in the other groups (*Figure 3F-H*). The trend of ATP and ROS was consistent with tumor size (*Figure 3I, J*), and the protein levels of C1qbp in the tumors were also reduced in the Pa28γ-silenced groups (*Figure 3K*). Collectively, these data suggest that PA28γ, which co-localizes with C1QBP in mitochondria, may involve in regulating mitochondrial morphology and function.

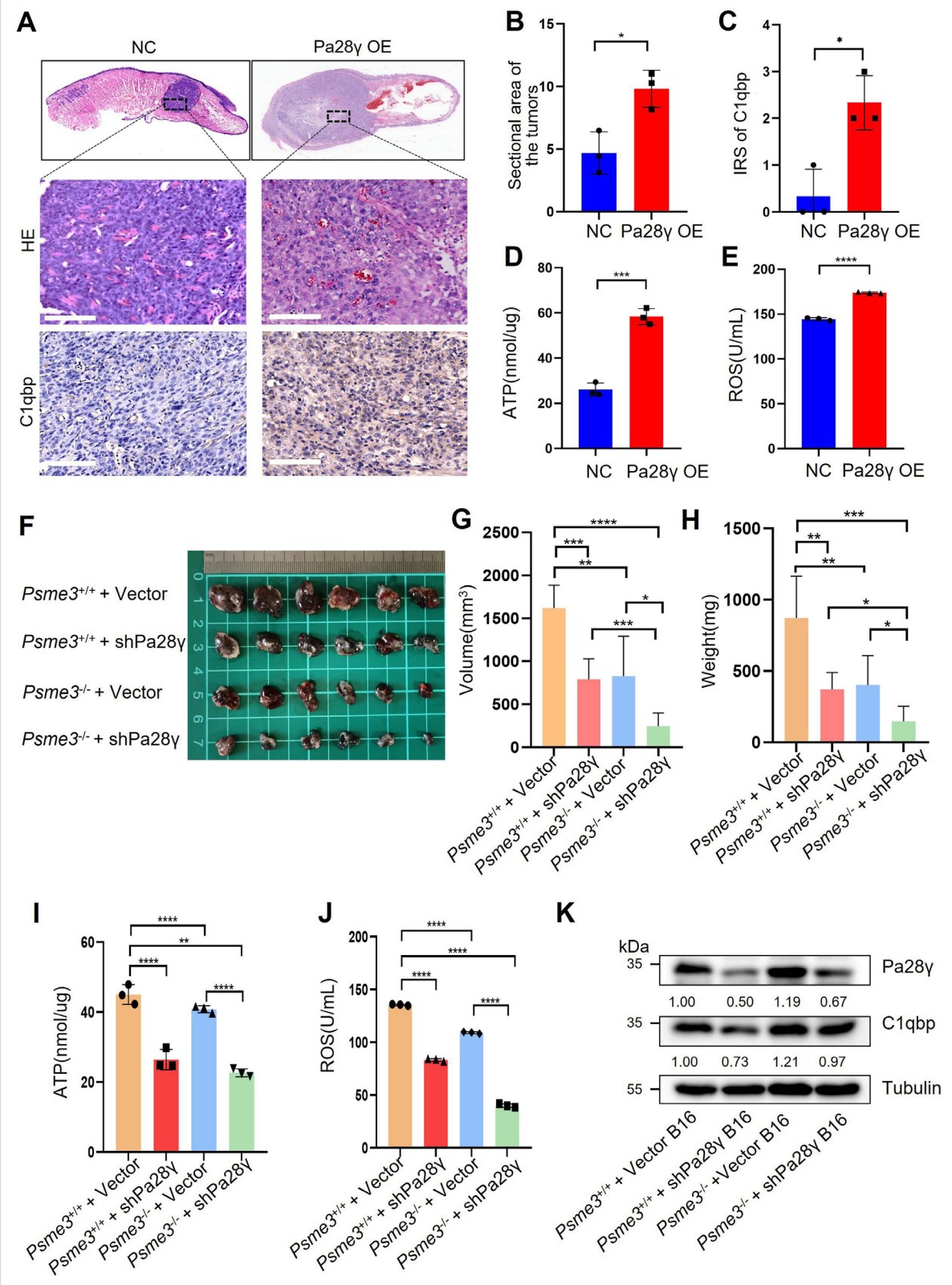

**Figure 3.** Proteasome activator 28γ (PA28γ) affect mitochondrial functions in vivo. (**A**) Representative images of H&E staining and immunohistochemical (IHC) staining of C1qbp of tongue sections from mice (n=3, scale bar=100µm). (**B**) Quantification of the sectional area in tumors from the Pa28γ-overexpressing and control groups (n=3; the data are presented as the means ± SDs of three samples; *p<0.05). (**C**) Comparison of the immunoreactive scores (IRSs) of C1qbp antibody staining in the Pa28γ-overexpressing and control groups (n=3, the data are presented as the means ± SDs; *p<0.05).

*Figure 3 continued on next page*

*Figure 3 continued*

(**D**, **E**) Quantification of ATP production and ROS levels in tumors from the Pa28γ-overexpressing and control groups (n=3; the data are presented as the means ± SDs of three samples; *p<0.05, ***p<0.001, ****p<0.0001). (**F**) Images of the tumors in different groups at the endpoint (n=6, scale bar=1cm). (**G**-**J**) The volume, weight, ATP, and ROS of tumors in different groups at the endpoint (n=6; the data are presented as the means ± SDs of three independent experiments; *p<0.05, **p<0.01, ***p<0.001, ****p<0.0001). (**K**) Western blot analysis of tumors in different groups.

The online version of this article includes the following figure supplement(s) for figure 3:

**Figure supplement 1.** Proteasome activator 28γ (PA28γ) overexpression can upregulate C1QBP in vivo.

## PA28γ regulates mitochondrial OXPHOS by upregulating C1QBP

Due to the results of targeted metabolomics analysis by mass spectrometry (*Figure 4—figure supplement 1A, B*), it was found that PA28γ may affect the biological behavior and function of OSCC by influencing metabolism. Therefore, we explored the protein levels of the mitochondrial respiratory chain complex, C1QBP, and other mitochondrial functional proteins. We found that the levels of C1QBP, complex I and IV proteins, as well as OPA1, MFN1, and MFN2 proteins, were upregulated in PA28γ-overexpressing OSCC cells (*Figure 4A, B*, *Figure 4—figure supplement 1C, D*), suggesting that the effect of PA28γ on the levels of these mitochondrial fusion factors is mediated by C1QBP via a molecular mechanism that is currently unknown. These increases could be reversed by C1QBP silencing in PA28γ-overexpressing OSCC cells (*Figure 4C*).

Consistent with this, the OCRs of C1QBP-silenced PA28γ-overexpressing OSCC cells were markedly lower than those of control cells (*Figure 4D*, *Figure 4—figure supplement 1E-G*). Subsequently, we constructed a series of experiments to detect the biological behavior of control and PA28γ-overexpressing OSCC cells with or without C1QBP silencing. C1QBP knockdown significantly attenuated the migration, invasive, and proliferation capabilities previously augmented by PA28γ overexpression (*Figure 4E-H*). These data suggest that PA28γ enhances mitochondrial OXPHOS function through C1QBP.

## C1QBP expression was positively associated with PA28γ expression and was associated with a worse prognosis in patients with OSCC or SKCM

To determine the correlation between PA28γ and C1QBP, IHC staining was performed in the clinical cohort of oral mucosa carcinogenesis. The results showed that both C1QBP and PA28γ were notably upregulated during the progression of oral mucosa carcinogenesis (*Figure 5A, B*, *Figure 5—figure supplement 1A*), and the levels of PA28γ and C1QBP were positively related (*Figure 5C*). Similarly, both C1QBP and PA28γ were upregulated in metastatic OSCC tissues (*Figure 5D, E*, *Figure 5—figure supplement 1B*), and the levels of PA28γ and C1QBP were consistently positively correlated in OSCC cohort (*Figure 5F*). Notably, consistent with our previous findings on PA28γ (*Li et al., 2015*), C1QBP or combining PA28γ and C1QBP could be a negative predator in our multicenter OSCC clinical cohort (*Figure 5G, H*), TCGA HNSC database (*Figure 5I, J*), and TCGA SKCM database (*Figure 5—figure supplement 1C, D*).

## Discussion

Metabolic reprogramming is one of the hallmarks of malignant tumors and is related to the malignant biological behavior of tumors (*Faubert et al., 2020*; *Hanahan, 2022*; *Tsai et al., 2023*; *Xia et al., 2021*). The metabolic phenotype of tumor cells differs from that of normal cells and dynamically changes during tumor progression. OSCC is a common head and neck malignancy that is still a growing global health problem (*Johnson et al., 2020*). The development of OSCC is a complex and multifaceted process in which metabolic reprogramming appears to be important (*Liu et al., 2023*). However, the mechanism of metabolic reprogramming remains unclear. Excitingly, we found the evidence that PA28γ interacts with and stabilizes C1QBP. We speculate that aberrantly accumulated C1QBP enhances the function of mitochondrial OXPHOS and leads to the production of additional ATP and ROS by activating the expression and function of OPA1, MNF1, MFN2, and mitochondrial respiratory chain complex proteins. This process results in mitochondrial fusion and malignant tumor

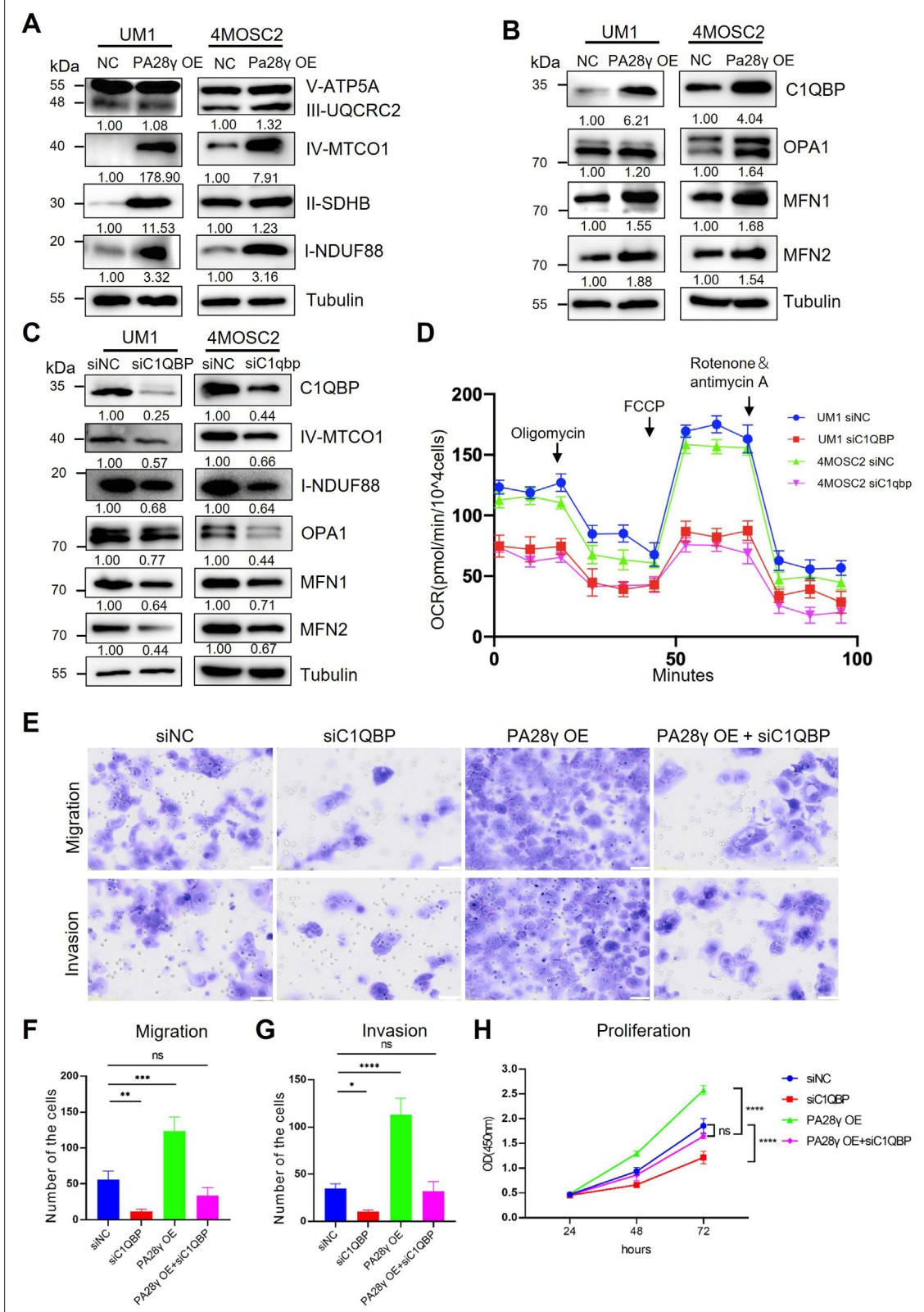

**Figure 4.** Proteasome activator 28γ (PA28γ) regulates mitochondrial oxidative phosphorylation (OXPHOS) and cellular biological behavior via C1QBP. (**A**, **B**) Western blot analysis of PA28γ-overexpressing and control UM1 and 4MOSC2 cells. (**C**) Western blot analysis of PA28γ-overexpressing oral squamous cell carcinoma (OSCC) cells transfected with siNC or siC1QBP. (**D**) OCRs of C1QBP-silenced and control PA28γ-overexpressing UM1 and 4MOSC2 cells (the data are presented as the means ± SDs of three independent experiments). (**E-H**) Cell migration, invasion and proliferation in control,

*Figure 4 continued on next page*

*Figure 4 continued*

C1QBP-silenced, PA28γ-overexpressing and PA28γ-overexpressing + C1QBP-silenced UM1 cells (scale bar=100μm, the data are presented as the means ± SDs of three independent experiments; *p<0.05, **p<0.01, ***p<0.001, ****p<0.0001).

The online version of this article includes the following figure supplement(s) for figure 4:

**Figure supplement 1.** Proteasome activator 28γ (PA28γ) and C1QBP are involved in oxidative phosphorylation (OXPHOS) and cellular biological behavior.

progression (*Figure 6*). Our study highlights the underlying mechanism by which PA28γ participates in the regulation of OXPHOS by upregulating C1QBP in OSCC.

C1QBP is also called gClqR, p32, p33, and HABP1, can be distributed inside cells, on the cell surface, and can also be secreted extracellularly (*Egusquiza-Alvarez and Robles-Flores, 2022*). Within cells, C1QBP is primarily located in the mitochondria. The key functions of mitochondrial C1QBP include maintaining OXPHOS, supplying energy to cells, and achieving homeostasis of mitochondrial metabolism (*Ghebrehiwet et al., 2019*; *Wang et al., 2022*). C1QBP is highly expressed in cells that require a large amount of energy, such as the heart, skeletal muscle, testis, ovary, small intestine, and colon (*Feichtinger et al., 2017*). Sufficient energy is essential for cancer development, and C1QBP promotes malignant behaviors such as tumor invasion and metastasis by enhancing OXPHOS function in various cancers (*Raschdorf et al., 2021*; *Wang et al., 2022*). However, there are no reports on the expression and functions of C1QBP in OSCC. In our study, we found the evidence that PA28γ interacts with and stabilizes C1QBP. As a proteasome activator, PA28γ usually affects the development of tumors by regulating the stability of various important proteins. For example, PA28γ can promote the degradation of p53, p21 leading to cancer progression (*Liu et al., 2010*). In addition, PA28γ can also play as a non-degradome role in tumor angiogenesis. For example, PA28γ can regulate the activation of NF-κB to promote the secretion of IL-6 and CCL2 in OSCC cells, thus promoting the angiogenesis of endothelial cells (*Liu et al., 2018*). However, the function of mitochondrial PA28γ may differ from that of nuclear PA28γ (*Mao et al., 2008*). Our study reveals that PA28γ interacts with C1QBP and stabilizes C1QBP at the protein level. Therefore, we speculate that the binding sites of PA28γ and C1QBP may mask the specific post-translational modification sites of C1QBP and inhibit its degradation. For example, the transcriptional coactivator p300 interacts with Smad7 by acetylating two lysine residues in its N-terminus, which stabilize Smad7 and protect it from TGFβ-induced degradation (*Grönroos et al., 2002*). Overall, the specific mechanism involved in the interaction between PA28γ and C1QBP still needs further exploration.

Considering that C1QBP is a vital protein for maintaining mitochondrial metabolism (*Tian et al., 2023*), we further investigated the function underlying the malignant progression of OSCC through the interaction between PA28γ and C1QBP in vitro and in vivo. PA28γ and C1QBP colocalized in the mitochondria, and the stabilization of C1QBP by PA28γ enhanced mitochondrial function and OXPHOS. This enhancement is crucial for ATP production, and ATP is the primary energy currency of cellular metabolism. Therefore, PA28γ's regulation of OXPHOS may impact cellular energy metabolism. PA28γ has been reported to activate the mTORC1 signaling pathway in hepatocellular carcinoma cells to promote glycolysis and inhibit OXPHOS (*Yao et al., 2021*). These findings contrast with our study, potentially due to organ heterogeneity or the complexity of metabolic reprogramming.

In the tissue microenvironment of oral lichen planus, a potentially malignant oral disorder, PA28γ in epithelial cells can regulate T cell differentiation (*Wang et al., 2024*), while PA28γ in CAFs is involved in the crosstalk between stromal cells and tumor cells (*Li et al., 2024*). This suggests that PA28γ may interact with the tumor immune microenvironment. Our phenotypically similar OSCC xenograft tumor models in both immunocompetent and nude mice indicate that the regulation of mitochondrial OXPHOS by PA28γ localized in tumor cell mitochondria does not entirely depend on the immune system (*Wang et al., 2019*).

Furthermore, our study reveals that PA28γ can regulate C1QBP and influence mitochondrial morphology and function by enhancing the expression of OPA1, MFN1, MFN2, and the mitochondrial respiratory complex. Mitochondrial fusion, crucial for oxidative metabolism and cell proliferation, is regulated by MFN1, MFN2, and OPA1. The first two fuse with the outer mitochondrial membrane, while the last fuses with the inner mitochondrial membrane (*Westermann, 2010*). OPA1 is an essential inner mitochondrial membrane protein with multiple functions, including regulating mitochondrial

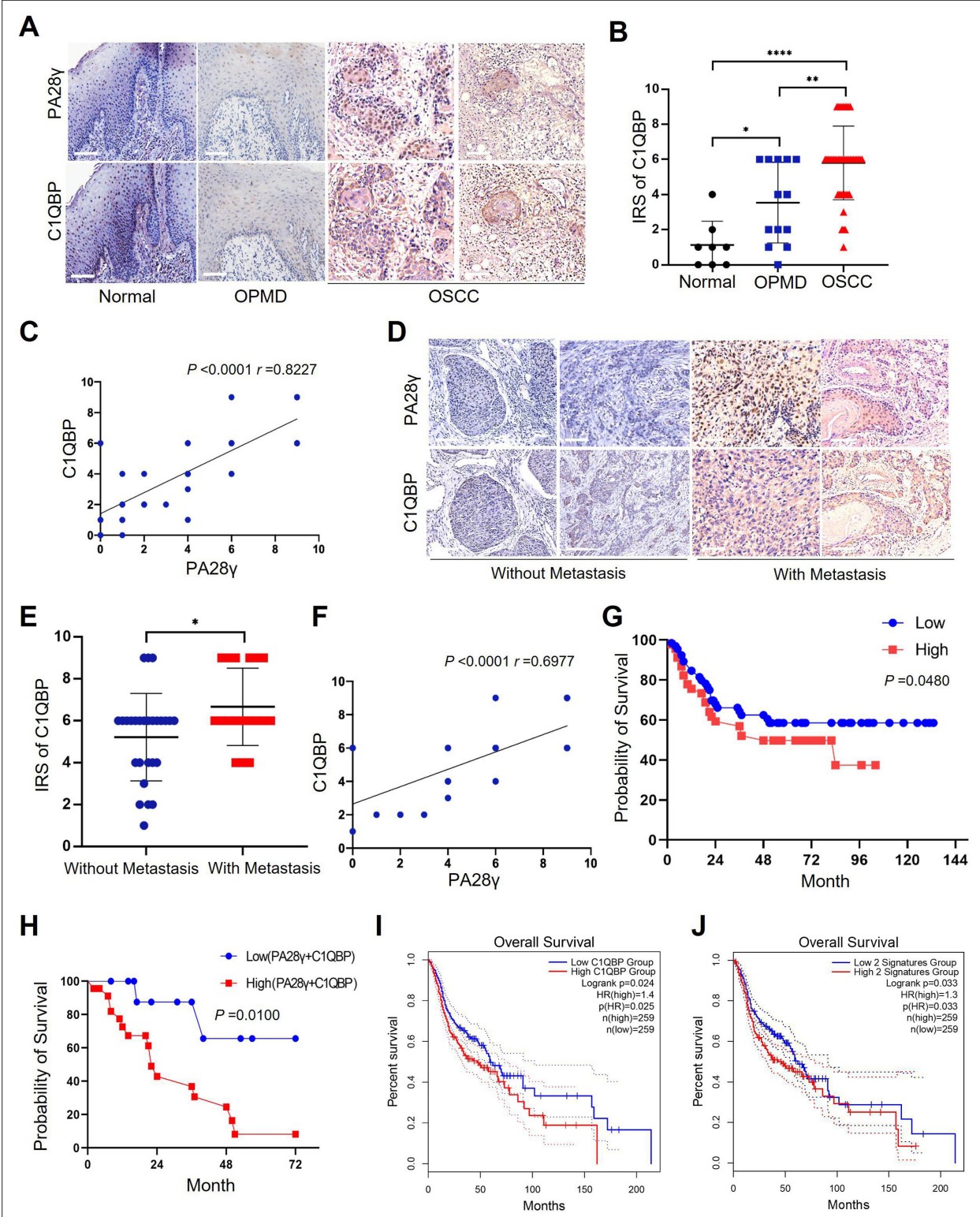

**Figure 5.** The correlation between proteasome activator 28γ (PA28γ) and C1QBP in the carcinogenesis and development of oral squamous cell carcinoma (OSCC). (**A**) Representative immunohistochemical (IHC) staining of PA28γ and C1QBP in normal (n=8), OPMD (n=13) and OSCC (n=45) samples (scale bar=200μm). (**B**) Comparison of the immunoreactive scores (IRSs) of C1QBP between the normal, OPMD, and OSCC groups (the data are presented as the means ± SDs; *p<0.05, **p<0.01, ****p<0.0001). (**C**) Spearman correlation analysis was used to test the correlation between PA28γ and C1QBP in normal, OPMD, and OSCC tissues (p<0.0001, r=0.8227). (**D**) Representative IHC staining of PA28γ and C1QBP in non-metastatic (n=27) and metastatic (n=18) OSCC patients (scale bar=200μm). (**E**) Comparison of the immunoreactive scores (IRSs) of C1QBP in the non-metastatic and metastatic

*Figure 5 continued on next page*

*Figure 5 continued*

OSCC groups (the data are presented as the means ± SDs; *p<0.05). (**F**) Spearman correlation analysis was used to test the correlation between PA28γ and C1QBP in OSCC tissues (p<0.0001, r=0.6977). (**G**) Kaplan–Meier analysis of the protein expression of C1QBP in our multicenter OSCC clinical cohort (n=295, P=0.0480). (**H**) Kaplan–Meier analysis of both low or high protein expression of C1QBP and PA28γ in our multicenter OSCC clinical cohort (n=295, p=0.0100). (**I**) Kaplan–Meier analysis of the protein expression of C1QBP in The Cancer Genome Atlas (TCGA) (HNSC) database (n=259, p=0.025). (**J**) Kaplan–Meier analysis of both low or high protein expression of C1QBP and PA28γ in TCGA HNSC database (n=259, p=0.033).

The online version of this article includes the following figure supplement(s) for figure 5:

**Figure supplement 1.** Proteasome activator 28γ (PA28γ) and C1QBP are involved in the development of tumor.

fusion, cristae morphology, mtDNA stability, and interacting with the mitochondrial respiratory chain to regulate OXPHOS (*Del Dotto et al., 2018*).

Finally, we analyzed the expression patterns and significance of PA28γ and C1QBP in a clinical cohort. PA28γ and C1QBP were negatively correlated with prognosis in the OSCC clinical cohort. Notably, some studies have shown that OXPHOS is upregulated in various tumors, with some advanced tumors preferring OXPHOS metabolically (*Ohshima and Morii, 2021*; *Qiu et al., 2023*). OXPHOS activity is associated with the recurrence of OSCC (*Noh et al., 2023*). Additionally, the enhancement of OXPHOS and ATP production by ROS proto-oncogene 1, which localizes to mitochondria in OSCC, promotes the OSCC invasion (*Noh et al., 2023*; *Grimm et al., 2014*). These findings are consistent with our results. Our study not only explains the mechanisms and signaling networks underlying the oncogenic potential of PA28γ but also contributes to understanding the possible mechanisms involved in OSCC metabolic reprogramming.

In summary, we found that PA28γ could interact with C1QBP and stabilize C1QBP. This enhances the function of OXPHOS, leading to increased production of ATP and ROS by increasing the expression

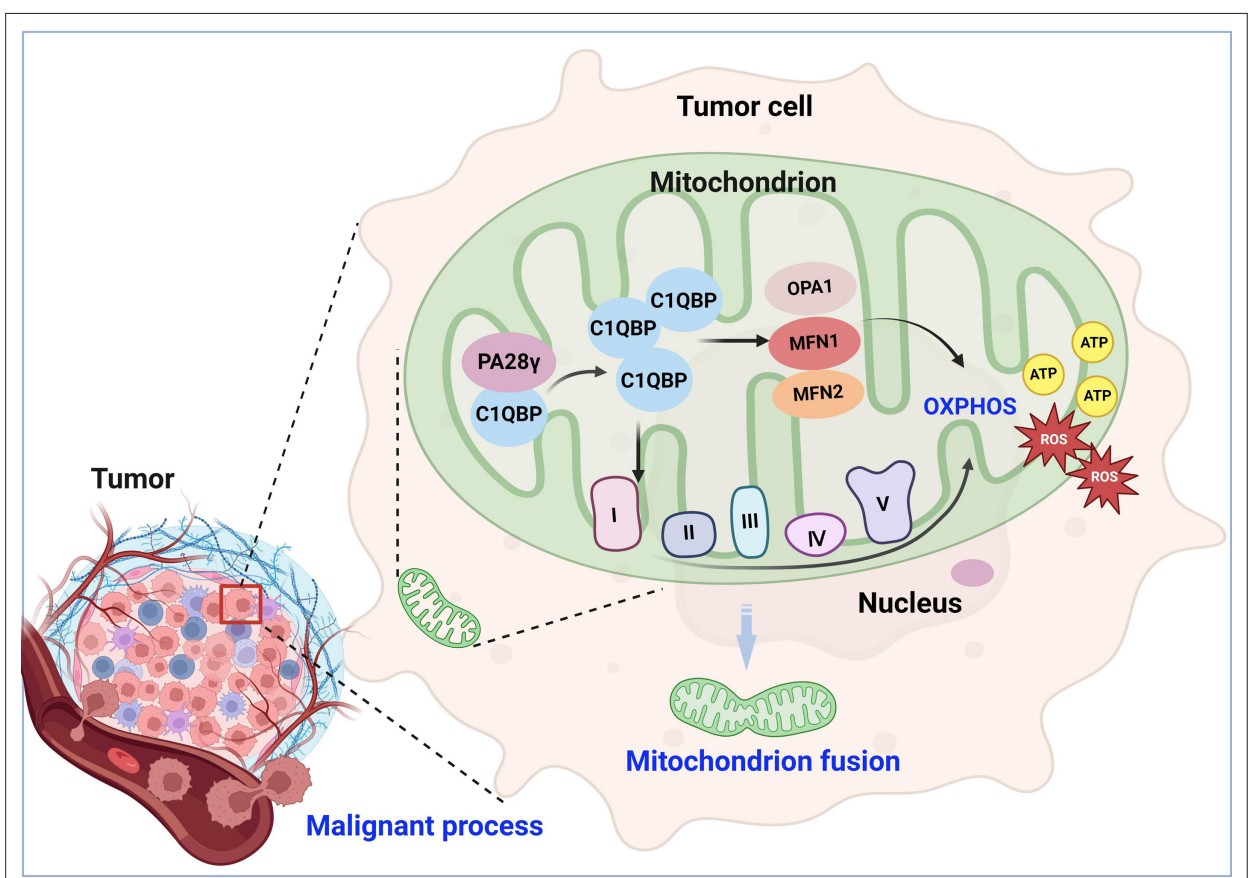

**Figure 6.** Molecular mechanism through which proteasome activator 28γ (PA28γ) interacts with C1QBP in the malignant progression of tumor. PA28γ can interact with and stabilize C1QBP, which can activate the expression and function of OPA1, MNF1, MFN2, and mitochondrial respiratory chain complex proteins, resulting in enhanced mitochondrial oxidative phosphorylation (OXPHOS), mitochondrial fusion, and malignant tumor progression.

and function of OPA1, MNF1, MFN2, and the mitochondrial respiratory chain complex. This process promotes mitochondrial fusion and malignant tumor progression. The expression of PA28γ and C1QBP was increased and positively correlated in OSCC. High expression of PA28γ and C1QBP was correlated with poor prognosis in OSCC patients, indicating that both proteins, in particular their molecular interactions, could serve as potential targets for the treatment and prognosis of OSCC.

## Materials and methods

### Patients and follow-up

Our study included four cohorts. The first cohort included eight normal controls, 13 patients with oral potentially malignant disorder (OPMD), and 45 patients with OSCC (*Appendix 1—table 1*). These patients visited the West China Hospital of Stomatology in 2021. The normal tissues were obtained from patients who underwent maxillofacial plastic surgery. The second cohort for survival analysis included 295 patients diagnosed with primary OSCC tumors who visited West China Hospital of Stomatology, Peking University Hospital of Stomatology, and Guangdong Provincial Stomatological Hospital from 2005 to 2009 (*Appendix 1—table 2*). None of the patients had cancer in other organs. These patients received regular follow-up for 1–133 months after the operation. The protocol of this study was approved by the ethics committee of the West China Hospital of Stomatology (Approval number: WCHSIRB-D-2020–046). All patients included in the study had signed informed consent forms and agreed to use their information for academic exchange and publication while protecting their privacy. The third cohort, comprising 518 patients with head and neck squamous cell carcinoma (HNSCC), was obtained from The Cancer Genome Atlas (TCGA) database (*Appendix 1—table 3*). The last cohort, comprising 458 patients with skin cutaneous melanoma (SKCM), was obtained from TCGA database (*Appendix 1—table 4*).

### Laboratory experiments

All animal studies were approved by the Animal Care and Use Committee for the State Key Laboratory of Oral Diseases (Approval number: WCHSIRB-D-2022–032) in compliance with the Guide for the updated Animal Research: Reporting of In Vivo Experiments (ARRIVE) 2.0 guidelines. The sample size for animal experiments is estimated based on existing research and empirical data, minimizing animal use while ensuring experimental validity. Other methods are detailed in the Appendix, including cell culture and cell-related experiments, animal models, mitochondrial function detection, western blot/immunoprecipitation assays, hematoxylin and eosin (H&E) staining, and immunohistochemistry.

### Statistical analysis

The data were analyzed using GraphPad Prism software version 8.0. The statistical data are expressed as the mean ± standard deviation (SD). Chi-square and Fisher exact tests, Spearman correlation tests, Wilcoxon rank-sum tests, one-way ANOVA, and Student's t-tests were used to analyze the data. The Kaplan–Meier method was used for survival analysis, and the log-rank test was used to evaluate the prognostic value of C1QBP in OSCC patients. The data are shown as the mean ± standard error, and a value of $p < 0.05$ was considered to indicate statistical significance: $p < 0.05^*$, $p < 0.01^{**}$, $p < 0.001^{***}$, and $p < 0.0001^{****}$. Each experiment was repeated three times.

## Acknowledgements

This work was supported by grants from the National Natural Science Foundation of China (82472686 and 82273320 to J Li, 82270986 and 82470983 to X Zeng, 82201074 to J Wang), and the Province Natural Science Foundation of Sichuan (23NSFSC0145 to J Li). The authors thank Zhijun Sun (School and Hospital of Stomatology, Wuhan University) for providing the cells needed for the experiment and Ning Ji (State Key Laboratory of Oral Diseases, Sichuan University) for excellent assistance with microscopic imaging.

## Additional information

### Funding

| Funder | Grant reference number | Author |
|---|---|---|
| National Natural Science Foundation of China | 82472686 | Jing Li |
| National Natural Science Foundation of China | 82270986 | Xin Zeng |
| National Natural Science Foundation of China | 82201074 | Jiongke Wang |
| Province Natural Science Foundation of Sichuan | 23NSFSC0145 | Jing Li |
| National Natural Science Foundation of China | 82273320 | Jing Li |
| National Natural Science Foundation of China | 82470983 | Xin Zeng |

The funders had no role in study design, data collection and interpretation, or the decision to submit the work for publication.

### Author contributions

Jiongke Wang, Xin Zeng, Jing Li, Conceptualization, Data curation, Formal analysis, Funding acquisition, Writing – original draft, Writing – review and editing; Yujie Shi, Conceptualization, Data curation, Formal analysis, Writing – original draft, Writing – review and editing; Ying Wang, Yingqiang Shen, Huan Liu, Data curation; Silu Sun, Yimei Wang, Formal analysis; Xikun Zhou, Yu Zhou, Qianming Chen, Conceptualization, Writing – review and editing

### Author ORCIDs

Jiongke Wang http://orcid.org/0000-0003-0326-6376
Xikun Zhou https://orcid.org/0000-0001-6768-2382
Jing Li https://orcid.org/0000-0001-5173-0781

### Ethics

The protocol of this study was approved by the ethics committee of the West China Hospital of Stomatology (Approval number: WCHSIRB-D-2020-046). All patients included in the study had signed informed consent forms and agreed to use their information for academic exchange and publication while protecting their privacy.

All animal studies were approved by the Animal Care and Use Committee for the State Key Laboratory of Oral Diseases (Approval number: WCHSIRB-D-2022-032) in compliance with the Guide for the updated Animal Research: Reporting of In Vivo Experiments (ARRIVE) 2.0 guidelines.

Reviewer #2 (Public review): https://doi.org/10.7554/eLife.101244.5.sa1
Author response https://doi.org/10.7554/eLife.101244.5.sa2

## Additional files

### Supplementary files

MDAR checklist

### Data availability

All data generated or analysed during this study are included in the manuscript and source data.

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

# Appendix 1

## Cell culture

The cell lines used in this study were not included in the list of commonly misidentified cell lines maintained by the International Cell Line Authentication Committee and free from mycoplasma contamination. Human OSCC cell lines (UM1, HSC-3, and HN12), human renal epithelial cell line (293T), and mouse melanoma cell line (B16) were obtained from the State Key Laboratory of Oral Diseases, whose identities had been authenticated through STR profiling by a third-party testing agency. UM1 and HSC-3 are tongue squamous cell carcinoma cell lines with different invasion and migration capabilities (*Nakayama et al., 1998*; *Momose et al., 1989*). UM1 cells are derived from primary tongue squamous cell carcinoma, while HSC-3 and HN12 cells are derived from metastatic cervical lymph nodes. The mouse OSCC line 4MOSC2 (gifted by J. Silvio Gutkind from the University of California, San Diego with Materials Transfer Agreement [MTA]: sd-2017-202) (*Wang et al., 2019*) was cultured in keratinocyte media (Invitrogen, USA) supplemented with Defined Keratinocyte-SFM Growth Supplement (Invitrogen, USA), 5 ng/mL EGF recombinant mouse protein (Invitrogen, USA) and cholera toxin (Sigma, USA) at 37°C and 5% $CO_2$. The mouse melanoma cell line (B16) was cultured in RPMI 1640 medium (HyClone, USA) supplemented with 10% FBS and 1% penicillin–streptomycin at 37°C and 5% $CO_2$. Human OSCC cell lines (UM1, HSC-3, and $HN1_2$) and a human renal epithelial cell line (293T) were maintained in DMEM (Sigma, USA) supplemented with 10% fetal bovine serum (HyClone, USA) at 37°C and 5% CO2, and the medium was changed every two days. The cells were passaged at 90% confluence and seeded at 30-40% confluence for maintenance of optimal proliferation conditions.

## Cell transfection and infection

Six-well plates were used to culture UM1, HSC-3, HN12, and 293T cells, and transfection was carried out when the cells were 70-80% confluent. Serum-free medium was used before transfection reagents were added to the corresponding plates. siRNAs were transiently transfected with Lipofectamine RNAiMAX Transfection Reagent (Thermo Fisher Scientific, USA), and the plasmids were transiently transfected with Lipofectamine 2000 (Invitrogen, USA). After being cultured for 48-72 hr, the cells were seeded. siRNAs for C1QBP were purchased from RiboBio (Guangzhou, China).

## Plasmids

Plasmids (pcDNA3.1-Myc, pcDNA3.1-Flag, pcDNA3.1-Flag-PA28γ, pcDNA3.1-Myc-PA28γ, pcDNA3.1-Flag-C1QBP, pcDNA3.1-Flag-C1QBP-T76A, and pcDNA3.1-Flag-C1QBP-G78N) and C1QBP-related plasmids were synthesized by Genewiz (Suzhou, China). The full-length C1QBP plasmid was named C1QBP-WT or C1QBP-Full. The C1QBP truncation plasmids pcDNA3.1-Flag-C1QBP 1-167, pcDNA3.1-Flag-C1QBP 94-282, and pcDNA3.1-Flag-C1QBP 168-282 had the amino acid 76-93 functional domain or amino acid 168-213 functional domain removed, respectively.

## Stable cell line generation

The Flag-PA28γ recombinant lentivirus PA28γ-OE and OE-vector were purchased from NeuronBiotech (Shanghai Genechem Co., Ltd.). UM1, HSC-3 and HN12 cells were infected with the PA28γ-OE or OE-vector. Selective culture medium containing puromycin was used to select cells stably expressing PA28γ-OE or vector controls. The expression of PA28γ was detected by Q-PCR and western blot analysis.

## Cell Invasion Assay

The in vitro invasion capability of the cells was measured using Transwell chambers coated with Matrigel (Corning, USA; 1:3 dilution with serum-free medium) in 24-well cell culture dishes. Matrigel (35 μL) was added to the upper chamber and incubated for 1 hr at 37°C to ensure that the Matrigel was solidified. Cells were seeded in 200 μl of FBS-free cell culture medium at a density of $5\times10^4$ cells/ml in the upper chamber, and the lower chamber was filled with 500 μL of culture medium supplemented with 10% FBS, which acted as a chemoattractant. Cell culture dishes with Transwell chambers were then incubated at 37°C and 5% $CO_2$ for 24 hr to allow the cells to invade. At the end of the incubation, the cells on the upper side of the Matrigel-coated filter were removed by wiping with a cotton swab. Crystal violet solution was used to stain invaded cells. The cells were counted under an inverted light microscope (Olympus, Japan). Cells in five randomized fields of view at ×200 magnification were counted and are presented as the average number of cells per field of view. The experiment was replicated three times.

## Cell migration assay

The assay was similar to the in vitro invasion assay except that no Matrigel was added to the chamber. The cell concentration in the upper chambers was $5×10^5$ /mL, and the cells were added to a 200 µL volume. Cells in five randomized fields of view at ×200 magnification were counted and are presented as the average number of cells per field of view. The experiment was replicated three times.

## Cell proliferation assay

After 24 hr of cell transfection, the cells were seeded into a 96-well plate at a density of 5000 cells per well. The culture medium was removed after 24 hr and 48 hr, respectively. The cell proliferation assay reagent CCK-8 (Beyotime, China) was added under dark conditions, and the ratio of CCK-8 reagent to complete culture medium was 1:9. After incubation in the dark for 1.5 hr, the optical density (OD) of each well was measured under 450 nm light.

## Proximity ligation assay

It was detected using Duolink PLA Kit (Sigma-Aldrich, USA) according to the manufacturer's instructions. The antibodies were PA28γ (1:1000, Cell Signaling Technology, USA), C1QBP (1:400, Proteintech, China), $IgG_M$ (1:200, Beyotime, China) and $IgG_R$ (1:200, Beyotime, China).

## Cell immunofluorescence assays

The cells were added to a 24-well plate ($1×10^5$ cells per well) with a cover glass at the bottom overnight at 37°C. The cells were fixed with 4% paraformaldehyde for 30 min and then treated with 0.5% Triton X-100 solution for 15 min. Then, the samples were incubated with Tomm20 (1:500, BD Biosciences, USA) and C1QBP (1:400, Abcam, USA) antibodies for 16 hr at 4°C. The next day, TRITC (1:500, Invitrogen, USA) and Alexa Fluor 594 (1:500, Invitrogen, USA) conjugated secondary antibodies were added, and the cells were incubated for 45 min. The cell nuclei were stained with DAPI (Beyotime, China) and observed under a fluorescence microscope. For labeling of mitochondria, the mitochondrial fluorescent dye MitoTracker Deep Red FM (Invitrogen, USA) was diluted to 25 nM in culture medium. ImageJ was used to measure the area of mitochondria in the fluorescence micrographs. Mitochondrial Network Analysis (MiNA, available at https://github.com/ScienceToolkit/MiNA; *ScienceToolkit, 2021*) an ImageJ-based tool, was used to calculate the mean branch length of mitochondria in fluorescence micrographs.

## Transmission electron microscopy

The cells were fixed in 4% glutaraldehyde (pH = 7.4) and stored in a refrigerator at 4°C. The samples were subsequently sent to Chengdu Lilai Biotechnology Co., Ltd., for osmic acid fixation, neutral resin embedding, and preparation of ultrathin sections. The mitochondrial morphology of the cells was observed using transmission electron microscopy.

## Allogeneic orthotopic transplantation tumor mouse model

All the experiments were conducted in accordance with the guidelines outlined in the Principles of Laboratory Animal Care (NIH) and were approved by the Ethics Committee of West China Hospital of Stomatology, Sichuan University (WCHSIRB-D-2022–023).

Six 8-week-old C57BL/6 mice were used as experimental animals and were purchased from Chengdu Dossy Experimental Animals Co., Ltd. Submucous injections into the tongue of each mouse were performed. A total of $1×10^6$ control or PA28γ-overexpressing 4MOSC2 cells were randomly injected into the mice subcutaneously, and the mice were divided into a control group (PA28γ NC, n=3) and an experimental group (PA28γ OE, n=3). Ten days later, carbon dioxide euthanasia was carried out. The collected tongue was cut along the midline of the tumor, and half of the tissue was fixed in 4% paraformaldehyde solution for 48 hr. After dehydration and embedding, subsequent hematoxylin and eosin (HE) staining was carried out, and the other half of the isolated tumor tissue was frozen at –80°C for ATP and ROS analysis.

## Subcutaneous transplantation tumor mouse model

A total of 24 C57BL/6 mice aged 6–8 weeks were used as experimental animals, of which 12 were *Psme3* whole-body knockout mice, with the other 12 being the same litter. All mice were obtained from our research group. Submucous injections into the back of each random grouping of mouse

were performed. A total of $1×10^6$ cells (100 μL) were injected subcutaneously, and mice were divided into the following four groups based on different genotypes and cell lines: *Psme3* wild-type mice injected with control B16 cells (*Psme3*$^{+/+}$ + Vector, n=6), *Psme3* wild-type mice injected with Pa28γ knockdown B16 cells (*Psme3*$^{+/+}$ + shPa28γ, n=6), *Psme3* whole-body knockout mice injected with control B16 cells (*Psme3*$^{-/-}$ + Vector, n=6) and *Psme3* whole-body knockout mice injected with Pa28γ knockdown B16 cells (*Psme3*$^{-/-}$ + shPa28γ, n=6). On the seventh day after injection of tumor cells, the formation of subcutaneous tumors on the back was visible. On the 15th day, the mice were euthanized with carbon dioxide, and the back tumor tissue was collected. The tumor volume was calculated as (length × width$^2$)/2 after excision. Each tumor was half placed in 4% paraformaldehyde to prepare paraffin sections and half stored in the –80°C for the ATP and ROS analysis.

## Mitochondrial function detection

Mitochondrial OXPHOS function was detected using a Seahorse XFe24 Analyzer (Agilent, USA) and Seahorse XF Cell Mito Stress Test Kit (Agilent, USA) according to the manufacturer's instructions. Cells were seeded in an XF24-well cell culture plate at a density of 10,000 cells/well for 24 hr at 37°C and 5% CO2. The medium was changed to 500 μL of Seahorse XF DMEM (pH 7.4), and the cells were incubated at 37°C without 1.5 μM oligomycin, 2 μM carbonyl cyanide-4-(trifluoromethoxy) phenylhydrazone (FCCP), or 0.5 μM rotenone and antimycin A to test ATP-linked respiration, the maximal OCR, and non-mitochondrial oxygen consumption, respectively. All OCRs were normalized according to the cell number.

For ROS production, cells were seeded into a 96-well plate ($1×10^4$ cells per well) overnight in the dark at 37°C. Then, 100 μL of ROS dye-containing culture medium [the ROS fluorescent probe DHE (KeyGEN, China) at a concentration of 100 μM] was added to each well, and the mixture was incubated for 45 min. After the cells were washed twice with PBS, the fluorescence intensity was measured at an excitation wavelength of 518 nm and an emission wavelength of 605 nm. ROS levels in tumors were calculated using a ROS detection kit (Beyotime, China) following the manufacturer's protocol. Ten milligrams of tissue from each group were cut, and the tumors in each group were placed in the same tube. Then, the tumors were homogenized into tissue homogenates using a homogenizer at 4°C in 1 mL of lysis solution, after which the protein concentration was determined via the BCA method after ultrasonic lysis. Solutions for blank, standard, control, and test tubes were prepared according to the protocol. The tubes were kept at 37°C for 2 min, after which 2 mL of developer was added. The solutions were mixed well and left at room temperature for 20 min before being measured at 550 nm in a microplate reader.

The ATP production of tumors was calculated using an ATP detection kit (KeyGEN, China) according to the manufacturer's protocol. Ten milligrams of tissue homogenate from each tumor were prepared using a homogenizer at 4°C in 100 μL of lysis solution, after which the protein concentration was determined via the BCA method after ultrasonic lysis. An ATP standard solution was prepared with lysis solution to construct standard curves at concentrations of 0.01, 0.03, 0.1, 0.3, 1, 3, and 10 μM. Then, 100 μL of detection solution was added to 96-well plates and kept at room temperature for 5 min. Then, 20 μL of standard sample or tumor sample was added to the wells. The spontaneous fluorescence intensity of each well was tested, and ATP production was calculated according to standard curves.

## RNA extraction and RT–qPCR analysis

Total RNA was extracted using TRIzol Reagent (Invitrogen, USA) according to the manufacturer's instructions. The primers for *PSME3*, *C1QBP*, and *ACTB* were purchased from RiboBio (Guangzhou, China). A FastStart Universal SYBR-Green Master Mix kit was purchased from Roche (Mannheim, Germany). qPCR for the detection of *PSME3* and *C1QBP* mRNA expression levels was performed in an ABI 7500 Real-Time PCR system (ABI, Foster City, USA). Relative mRNA expression was normalized to that of *ACTB*. The specific primer sequences for the genes used were as follows:

*ACTB* forward: 5'- CACCATTGGCAATGAGCGGTTC –3',
reverse: 5'- AGGTCTTTGCGGATGTCCACGT –3';
*PSME3* forward: 5'- AAGGTTGATTCTTTCAGGGAGC –3',
reverse: 5'-AGTGGATCTGAGTTAGGTCATGG –3';
*C1QBP* forward: 5'- CACACCGACGGAGACAAAG –3',
reverse: 5'- GGGAGGGGTTTTATGCTTCTGAAT –3';

## Western blot assay

The cells were lysed in lysis buffer (1% NP-40 supplemented with a complete protease inhibitor tablet (self-formulated)) on ice. Protein extracts (30 µg) were separated by a 10% SDS–PAGE gel and transferred to 0.45 µm polyvinylidene difluoride (PVDF) membranes (Millipore, USA). After blocking with 5% bovine serum albumin (BSA) for 1 hr, the membranes were incubated with the following primary antibodies overnight at 4°C: α-Tubulin (Cell Signaling Technology, USA, 1:5000 dilution), Flag (Cell Signaling Technology, USA, 1:2000 dilution), Myc (Cell Signaling Technology, USA, 1:1000 dilution), C1QBP (Cell Signaling Technology, USA, 1:1000 dilution), PA28γ (Cell Signaling Technology, USA, 1:1000 dilution), OPA1 (Cell Signaling Technology, USA, 1:1000 dilution), Mitofusin-1 (Cell Signaling Technology, USA, 1:1000 dilution), Mitofusin-2 (Cell Signaling Technology, USA, 1:1000 dilution) NDUFB88 (Abcam, USA, 1:1000 dilution), SDHB (Abcam, USA, 1:1000 dilution), MTCO1 (Abcam, USA, 1:1000 dilution), and Total OXPHOS Rodent WB Antibody Cocktail (Abcam, USA, 1:1000 dilution). Afterwards, the membranes were incubated with either an anti-goat, anti-mouse, or anti-rabbit HRP-conjugated secondary antibody (Sigma–Aldrich, Australia, 1:3000 dilution) for 1 hr. After incubation with a chemiluminescence substrate, images were taken with an ImageReader LAS-2000 (Fujifilm, Japan), and the proteins were analyzed with ImageJ software.

## Immunoprecipitation assay

The cells were lysed in lysis buffer 1% NP-40 supplemented with a complete protease inhibitor tablet (self-formulated) on ice, and 20 µL of cell lysate was used as the input. The cell lysates were coincubated with antibodies at 4°C overnight. The protein G agarose was washed and added to a 1 mL sample at a ratio of 20 µl. The sample was shaken on a horizontal shaker for 4 hr, after which the deposit was collected for western blotting.

## HE staining analysis

The tissues were immersed in 4% paraformaldehyde for 24 hr and washed overnight with running water. The tissue was then placed in an automatic dehydrator. After dehydration, the tissue was embedded in paraffin and sectioned at a thickness of 5 µm. Dewaxing and gradient alcohol hydration were conducted before the sections were counterstained with hematoxylin and eosin. After dehydration with an alcohol gradient, the sections were sealed with neutral gum.

## Immunohistochemistry and analysis

Immunohistochemistry (IHC) for PA28γ (Cell Signaling Technology, USA, 1:400 dilution) and C1QBP (Abcam, USA, 1:800 dilution) was performed on FFPE specimens after antigen retrieval with EDTA buffer (1 mM, pH 8.0), and the proteins were visualized by diaminobenzidine (DAB, Gene Tech, 1:50 dilution). The MVD was calculated as the average count of microvessels in four hot spots at high magnification (×400). The intensity of staining (0, no staining; 1, weakly stained; 2, moderately stained; 3, strongly stained) was assessed by an experienced pathologist without any knowledge of the clinical or pathological data, and the percentage of positive cells was noted. The staining proportions were scored as 0 (≤5%), 1 (5–33%), 2 (34–66%), or 3 (≥67%), and the total staining was expressed as a product of the two numbers (six levels: 1, 2, 3, 4, 6 and 9). For the statistical analysis of prognostic value in the OSCC cohort, the total staining scale was divided into two categories: 1 (staining scale ≤4) and 2 (staining scale >4).

## Bioinformatics

To identify the specific genes targeted by *PSME3* and *C1QBP* in OSCC, GeneMANIA was used. To determine the prognostic value of *C1QBP*, Gene Expression Profiling Interactive Analysis (GEPIA) (http://gepia.cancer-pku.cn) was used to analyze the survival of OSCC patients in the TCGA HNSC and SKCM databases.

## Prediction and analysis of protein interactions

### Protein sequence retrieval and structure prediction

The protein sequences of C1QBP and PA28γ were obtained from the AlphaFold Protein Structure Database. Structural predictions of the protein-protein interaction between C1QBP and PA28γ were conducted using AlphaFold 3. The plDDT (predicted local distance difference test) values were utilized to assess the confidence of the predicted models. Models with a plDDT score above 70 were considered confident, while those with a score above 90 were categorized as very high confidence. These values were annotated in the figures to indicate the reliability of the structural predictions.

## Protein preparation and structure optimization

The best-scoring model for the C1QBP-PA28γ interaction predicted by AlphaFold 3 was selected for further analysis. The model was imported into MOE 2022 (Molecular Operating Environment) software for protein preparation. This process included the removal of water molecules and other heteroatoms, followed by the addition of hydrogen atoms to the structure. This step was essential for optimizing the protein's 3D conformation and ensuring the correctness of the protonation states at physiological pH.

## Energy minimization and hydrogen bond prediction

The protein structure was subjected to energy minimization using the Amber10: EHT (Effective Hamiltonian Theory) force field, with R-field 1: 80 settings to refine the model's geometry. The minimization process was performed to optimize the protein's internal energy and ensure stable conformation, followed by calculation of hydrogen bond interactions. The interaction energies and hydrogen bonds were analyzed to identify potential binding sites and stabilize the predicted protein-protein complex.

**Appendix 1—table 1.** Baseline Characteristics of the Patients with Oral Squamous Cell Carcinoma in Cohort I.

| characteristic | N=45 Number (%) | P value |
|---|---|---|
| Gender | | |
| Male | 26 (57.8) | |
| Female | 19 (42.2) | 0.7553 |
| Age（year） | | |
| <60 | 11 (24.4) | |
| ≥60 | 34 (75.6) | 0.3428 |
| Tumor stage | | |
| T1-2 | 30 (66.7) | |
| T3-4 | 15 (33.3) | 0.0030 |
| Node stage | | |
| N0 | 27 (60.0) | |
| N1-3 | 18 (40.0) | 0.0217 |
| Clinical stage | | |
| I-II | 23 (51.1) | |
| III-IV | 22 (48.9) | 0.0045 |
| Differentiation degree | | |
| Low/Moderate | 11 (24.4) | |
| High | 34 (75.6) | 0.2651 |
| Recurrence | | |
| Yes | 26 (57.8) | |
| No | 19 (42.2) | 0.5900 |
| C1QBP expression | | |
| Low | 13 (28.9) | |
| High | 32 (71.1) | <0.05 |

**Appendix 1—table 2.** Baseline Characteristics of the Patients with Oral Squamous Cell Carcinoma in Cohort II.

| characteristic | N=295 Number (%) | P value |
|---|---|---|
| Gender | | |
| Male | 216 (73.2) | |
| Female | 79 (26.8) | 0.0798 |
| Age（year） | | |
| <60 | 126 (42.7) | |
| ≥60 | 169 (57.3) | 0.6825 |
| Smoking | | |
| Yes | 145 (49.2) | |
| No | 150 (50.8) | 0.9483 |
| Tumor stage | | |
| T1-2 | 219 (74.2) | |
| T3-4 | 76 (25.8) | 0.7181 |
| Node stage | | |
| N0 | 194 (65.8) | |
| N1-3 | 101 (34.2) | 0.5265 |
| Clinical stage | | |
| I-II | 155 (52.5) | |
| III-IV | 140 (47.5) | 0.9280 |
| Recurrence | | |
| Yes | 149 (50.5) | |
| No | 146 (49.5) | 0.1889 |
| C1QBP expression | | |
| Low | 132 (44.7) | |
| High | 163 (55.3) | <0.05 |

**Appendix 1—table 3.** Baseline characteristics of the patients with head and neck squamous cell carcinoma in The Cancer Genome Atlas (TCGA) cohort.

| Characteristic | C1QBP expression | | P-value |
|---|---|---|---|
| | Low (N=259, Number %) | High (N=259, Number %) | |
| Gender | | | |
| Male | 190 (73.3) | 193 (74.5) | |
| Female | 69 (26.7) | 66 (25.5) | 0.8414 |
| Age（year） | | | |
| <60 | 133 (51.4) | 100 (38.6) | |
| ≥60 | 126 (48.6) | 159 (61.4) | 0.0047 |
| Tumor stage* | | | |
| I-II | 57 (22.0) | 40 (15.4) | |
| III-IV | 161 (62.2) | 186 (71.8) | 0.0384 |
| Clinical stage* | | | |
| I-II | 66 (25.5) | 51 (19.7) | |
| III-IV | 189 (73.0) | 199 (76.8) | 0.1702 |

*Appendix 1—table 3 Continued on next page*

*Appendix 1—table 3 Continued*

| Characteristic | C1QBP expression | | |
| --- | --- | --- | --- |
| | Low (N=259, Number %) | High (N=259, Number %) | P-value |
| Histological grade* | | | |
| G1-2 | 174 (67.2) | 191 (73.7) | |
| G3-4 | 72 (27.8) | 60 (23.2) | 0.1876 |

*The information of some patients is missing.

**Appendix 1—table 4.** Baseline characteristics of the patients with skin cutaneous melanoma in The Cancer Genome Atlas (TCGA) cohort.

| Characteristic | C1QBP expression | | |
| --- | --- | --- | --- |
| | Low (N=229, Number %) | High (N=229, Number %) | P-value |
| Gender | | | |
| Male | 143 (62.4) | 145 (63.3) | |
| Female | 86 (37.6) | 84 (36.7) | 0.9230 |
| Age (year) | | | |
| <60 | 107 (46.7) | 102 (44.5) | |
| ≥60 | 122 (53.3) | 127 (55.5) | 0.7075 |
| Tumor stage* | | | |
| I-II | 77 (33.6) | 71 (31.0) | |
| III-IV | 117 (51.1) | 122 (53.3) | 0.6013 |
| Node stage* | | | |
| 0 | 106 (46.3) | 123 (53.7) | |
| 1–3 | 94 (41.0) | 81 (35.4) | 0.1599 |
| Metastasis stage* | | | |
| 0 | 205 (89.5) | 204 (89.1) | |
| 1 | 11 (0.05) | 11 (0.05) | 0.9999 |
| Clinical stage* | | | |
| I-II | 102 (44.5) | 118 (51.5) | |
| III-IV | 103 (45.0) | 87 (38.0) | 0.1373 |

*The information of some patients is missing.

