## [Editor Report · eLife Assessment]

This manuscript determines how PA28g, a proteasome regulator that is overexpressed in tumors, and C1QBP, a mitochondrial protein for maintaining oxidative phosphorylation that plays a role in tumor progression, interact in tumor cells to promote their growth, migration and invasion. Additional experiments and analyses that supported the theoretical models for the interaction have been performed in response to the reviews. The overall findings and conceptual framework are **important** and the evidence is **solid**. A logical extrapolation of this work is to test the C1QBP mutants using functional assays to determine whether the mutations can decrease the protein stability mediated by the interaction with PA28g.

---

## [Referee Report · Reviewer #2 (Public review)]

Summary:

The authors tried to determine how PA28g functions in oral squamous cell carcinoma (OSCC) cells. They hypothesized it may act through metabolic reprogramming in the mitochondria.

Strengths:

They found that the genes of PA28g and C1QBP are in an overlapping interaction network after an analysis of a genome database. They also found that the two proteins interact in coimmunoprecipitation and pull-down assays using the lysate from OSCC cells with or without expression of the exogenous genes. They used truncated C1QBP proteins to map the interaction site to the N-terminal 167 residues of C1QBP protein. They observed the levels of the two proteins are positively correlated in the cells. They provided evidence for the colocalization of the two proteins in the mitochondria and the effect on mitochondrial form and function in vitro and in vivo OSCC models, and the correlation of the protein expression with the prognosis of cancer patients.

Comments on revision:

The third revision added data from two point mutations of C1QBP that would disrupt a hydrogen bond network with PA28g protein. As one would expect from the structural models obtained with AlphaFold, the interaction between the two proteins as detected by co-immunoprecipitation of cell lysate was reduced by both mutations. Therefore, the theoretical models for the interaction were supported by the experimental data. Moving forward, the home run experiments would be to test the C1QBP mutants in functional assays to determine whether the mutations can decrease the protein stability afforded by the interaction with PA28g, which in turn decrease the effect of PA28g on mitochondria and tumor cells via C1QBP. Success of these experiments will conclude this manuscript that presents a novel finding for tumor cell biology which could be a launch pad for therapeutic intervention of tumor development.

---

## [Author Response]

The following is the authors’ response to the previous reviews

**Public Reviews:**

**Reviewer #2 (Public review):**
This manuscript determines how PA28g, a proteasome regulator that is overexpressed in tumors, and C1QBP, a mitochondrial protein for maintaining oxidative phosphorylation that plays a role in tumor progression, interact in tumor cells to promote their growth, migration and invasion. Evidence for the interaction and its impact on mitochondrial form and function was provided although it is not particularly strong.The revised manuscript corrected mislabeled data in figures and provides more details in figure legends. Misleading sentences and typos were corrected. However, key experiments that were suggested in previous reviews were not done, such as making point mutations to disrupt the protein interactions and assess the consequence on protein stability and function. Results from these experiments are critical to determine whether the major conclusions are fully supported by the data.The second revision of the manuscript included the proximity ligation data to support the PA28g-C1QBP interaction in cells. However, the method and data were not described in sufficient detail for readers to understand. The revision also includes the structural models of the PA28g-C1QBP complex predicted by AlphaFold. However, the method and data were not described with details for readers to understand how this structural modeling was done, what is the quality of the resulting models, and the physical nature of the protein-protein interaction such as what kind of the non-covalent interactions exist in the interface of the protein complexes. Furthermore, while the interactions mediated by the protein fragments were tested by pull-down experiments, the interactions mediated by the three residues were not tested by mutagenesis and pull-down experiments. In summary, the revision was improved, but further improvement is needed.

Thank you very much for your comments.

(1) Based on your suggestion, we predicted the possible interaction sites using AlphaFold 3 and found that mutations in amino acids 76 and 78 of C1QBP affect the interaction with PA28γ (Revised Appendix Figure 1J). Subsequently, pulldown experiment also found that after mutating the amino acids at the two aforementioned sites (T76A, G78N), C1QBP that could bind to PA28γ decreased (Revised Figure 1J). The above results confirm that PA28γ could interacts with C1QBP, in a manner dependent on the N-terminus of C1QBP. These findings are now included in the revised manuscript “In addition, we employed AlphaFold 3 to perform energy minimization and predict hydrogen bonds between the C1QBP N-terminus (amino acids 1-167) and the PA28γ protein interaction region. The results suggest that the T76 and G78 residues of C1QBP may be key contributors to the interaction. Consistently, coimmunoprecipitation analysis demonstrated that mutations at these sites (C1QBPT76A and C1QBPG78N) significantly reduced the binding ability to PA28γ (Fig. 1J and Appendix Fig. 1J)”, specifically in results section. We believe this additional validation strengthens the robustness of our findings.

(2) According to your suggestion, we have added a description of the results of PLA in the figure legend (Revised Figure 1C) and the method of PLA in the appendix file (Revised Appendix file, Part “Proximity Ligation Assay”). The revised text reads as follows: (C) PLA image of UM1 cells shows the interaction between C1QBP and PA28γ in both cytoplasm and nucleus (red fluorescence).

(3) In the light of your suggestion, we have enriched the description of AlphaFold 3 analysis in the appendix file (Revised Appendix file, Page 10-11). The revised text reads as follows:

“Prediction and Analysis of Protein Interactions

Protein Sequence Retrieval and Structure Prediction

The protein sequences of C1QBP and PA28γ were obtained from the AlphaFold Protein Structure Database. Structural predictions of the protein-protein interaction between C1QBP and PA28γ were conducted using AlphaFold 3. The plDDT (predicted local distance difference test) values were utilized to assess the confidence of the predicted models. Models with a plDDT score above 70 were considered confident, while those with a score above 90 were categorized as very high confidence. These values were annotated in the figures to indicate the reliability of the structural predictions.”

“Protein Preparation and Structure Optimization

The best-scored model for the C1QBP-PA28γ interaction predicted by AlphaFold 3 was selected for further analysis. The model was imported into MOE 2022 (Molecular Operating Environment) software for protein preparation. This process included the removal of water molecules and other heteroatoms, followed by the addition of hydrogen atoms to the structure. This step was essential for optimizing the protein’s 3D conformation and ensuring the correctness of the protonation states at physiological pH.”

“Energy Minimization and Hydrogen Bond Prediction

The protein structure was subjected to energy minimization using the Amber10: EHT (Effective Hamiltonian Theory) force field, with R-field 1: 80 settings to refine the model’s geometry. The minimization process was performed to optimize the protein’s internal energy and ensure stable conformation, followed by calculation of hydrogen bond interactions. The interaction energies and hydrogen bonds were analyzed to identify potential binding sites and stabilize the predicted protein-protein complex.”